# Localization and interaction of interlayer excitons in MoSe$_2$/WSe$_2$ heterobilayers

Hanlin Fang [1]✉, Qiaoling Lin[2], Yi Zhang [3], Joshua Thompson[4], Sanshui Xiao [2], Zhipei Sun [3], Ermin Malic [4], Saroj P. Dash [1] & Witlef Wieczorek [1]✉

Transition metal dichalcogenide (TMD) heterobilayers provide a versatile platform to explore unique excitonic physics via the properties of the constituent TMDs and external stimuli. Interlayer excitons (IXs) can form in TMD heterobilayers as delocalized or localized states. However, the localization of IX in different types of potential traps, the emergence of biexcitons in the high-excitation regime, and the impact of potential traps on biexciton formation have remained elusive. In our work, we observe two types of potential traps in a MoSe$_2$/WSe$_2$ heterobilayer, which result in significantly different emission behavior of IXs at different temperatures. We identify the origin of these traps as localized defect states and the moiré potential of the TMD heterobilayer. Furthermore, with strong excitation intensity, a superlinear emission behavior indicates the emergence of interlayer biexcitons, whose formation peaks at a specific temperature. Our work elucidates the different excitation and temperature regimes required for the formation of both localized and delocalized IX and biexcitons and, thus, contributes to a better understanding and application of the rich exciton physics in TMD heterostructures.

Transition metal dichalcogenide (TMD) heterostructures provide a versatile 2D material platform to explore unique excitonic phenomena[1,2] including the realization of hybridized excitons[3], excitonic Mott insulators[4], and excitonic Bose-Einstein condensation[5]. TMD heterobilayers with type-II band alignment can host interlayer excitons (IXs), which consist of an electron and a hole that after a fast charge transfer process are located in the different constituent TMD monolayers[6,7]. The dipolar exciton nature provides the IX states with a long exciton lifetime[8], high valley polarization degree[9], and high electric field tunability[10], paving the way to excitonic optoelectronic applications[11]. The properties of IXs are strongly affected by the twist angle of the heterostructure[12,13], which modulates the energy landscape in the material and forms a moiré potential. Stacked TMD monolayers with small twist angles (typically less than 2° or larger than 58°) result in a strong moiré effect, which is manifested by

characteristic emission features of moiré-trapped IXs[14–16]. Such IXs could serve as quantum emitter arrays[12], realize optical nonlinearities[17], and explore many-body physics[18]. Experimentally, the localization of excitons in the moiré potential has been demonstrated by time-resolved and angle-resolved photoemission spectroscopy[19,20], exciton transport measurements[21], and electron-beam-based hyperspectral imaging[22].

An exemplary TMD heterobilayer system is MoSe$_2$/WSe$_2$, where two types of features are ascribed to moiré excitons that have been observed in the photoluminescence (PL) spectrum[14,23–25]. The first feature is sharp emission lines with a linewidth of 100 μeV, which do not show a systematic change with twist angle[23]. The second feature is characterized by multi-peaked emission features, where the individual peaks have a linewidth on the order of 20 meV and their energy spacing depends on the twist angle, which is in agreement with quantized

[1]Department of Microtechnology and Nanoscience (MC2), Chalmers University of Technology, 41296 Gothenburg, Sweden. [2]Department of Electrical and Photonics Engineering, Technical University of Denmark, DK-2800 Kongens Lyngby, Denmark. [3]Department of Electronics and Nanoengineering and QTF Centre of Excellence, Aalto University, Espoo 02150, Finland. [4]Department of Physics, Philipps-Universität Marburg, 35037 Marburg, Germany. ✉e-mail: hanlin.fang@chalmers.se; witlef.wieczorek@chalmers.se

states in the moiré potential[14]. Notably, the moiré potential can enhance the interaction between excitons and in that way enables the formation of many-body excitonic states, such as biexcitons[26]. However, the transition and relation between these excitonic features as well as the emergence of interlayer biexcitons (IXXs) in the strong excitation regime have not been explored so far.

In our work, we observe the two typical IX emission features mentioned above in the same H-stacked MoSe$_2$/WSe$_2$ heterobilayer. We find that these two types of IXs dominate the PL emission spectrum at different temperature ranges, thereby extending previous studies[14,23]. We show that the two emission features can be explained by the presence of two types of potential traps, of which one is shallow (4 meV) and possibly related to defect potentials, and the other one is deep (27 meV) and attributed to the moiré potential. At large excitation intensity, the dipolar interaction between the IXs enables the observation of the signature of PL emission originating from IXXs. We find that the conversion between IX and IXX reaches a maximal value at a temperature of 30 K, which correlates with the temperature at which the IX emission associated with the shallow potential traps is quenched.

## Results and discussion
### Interlayer excitons in a TMD heterobilayer
Figure 1a shows an optical microscope image of our 2D heterostructure fabricated via the van der Waals pickup method (details in "Methods"). The MoSe$_2$/WSe$_2$ heterobilayer is encapsulated by thin-hBN flakes to reduce the inhomogeneous screening from the

substrate[27], enabling the observation of multiple moiré-related excitonic resonances[14]. The constituent monolayers are twisted with an angle of approximately 58° (see Supplementary Fig. S1), thus, forming a moiré superlattice that can host various exciton states (see Fig. 1b). Note that atomic reconstruction[28] would impact the moiré superlattice for small twist angles (about ≤1° or ≥59° in a MoSe$_2$/WSe$_2$ heterobilayer), which we expect not to be the case in our sample. The spin-orbit coupling (SOC) results in a sizeable energy level splitting of the conduction band of MoSe$_2$, permitting the formation of spin-conserved singlet (S) and spin-forbidden triplet (T) IX states as shown in Fig. 1c. Unlike TMD monolayers that host optically dark spin-flip excitons, the heterobilayer structure breaks the mirror symmetry of the material and brightens the T-IXs[29].

We first excite the heterobilayer in an intermediate excitation regime (pump power of 66 μW) by a continuous wave (cw) 730 nm (corresponding to 1.7 eV) laser near-resonant to the intralayer exciton of WSe$_2$. Two interlayer excitonic states (peak at 1.4 eV and 1.425 eV, labeled as T and S, respectively) are observed in the PL emission of the heterobilayer (Fig. 1d). The two peaks are further characterized by polarization-dependent PL measurements, whereby the S-IX shows the expected cross-polarized and the T-IX co-polarized emission with the pump (see Supplementary Fig. S2). The observed energy splitting of 25 meV together with the measured valley polarization of the two peaks are in agreement with previous measurements of the SOC-induced singlet and triplet interlayer excitons in H-stacked MoSe$_2$/WSe$_2$ heterobilayers[9,30].

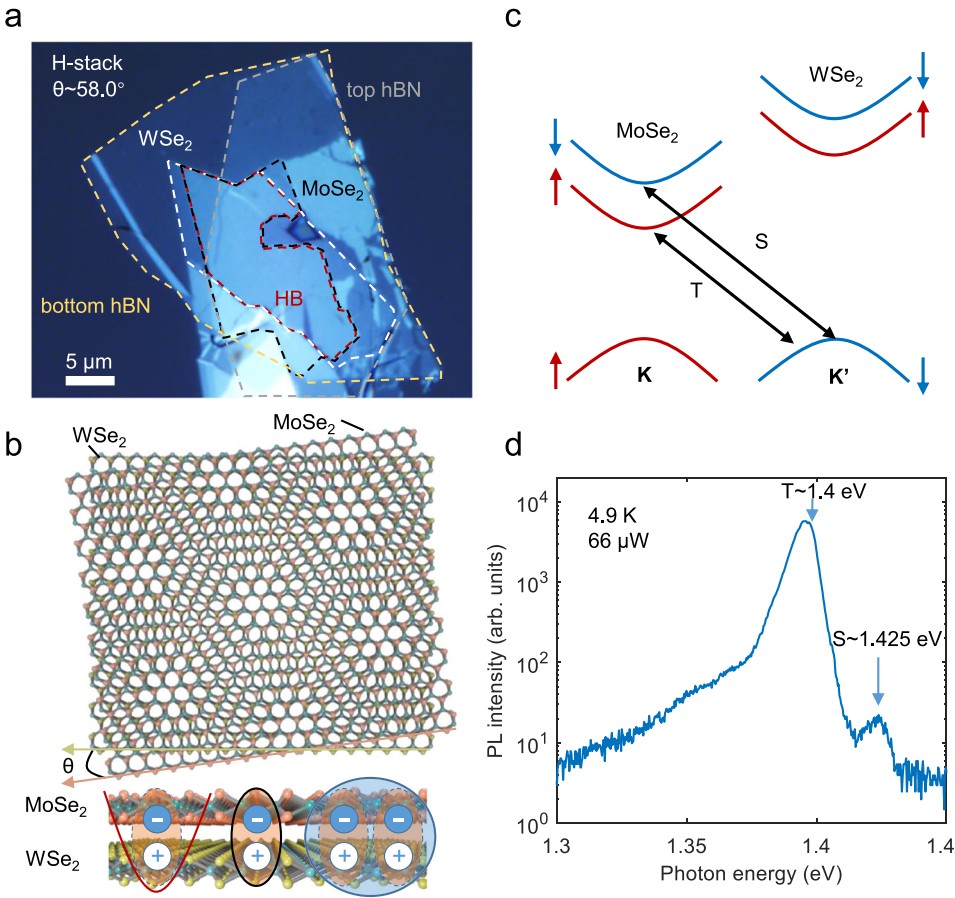

**Fig. 1 | Interlayer excitons in an hBN encapsulated H-stacked MoSe$_2$/WSe$_2$ heterobilayer sample. a** Bright-field optical image of the heterostructure on a fused silica substrate. HB (dashed red) is the MoSe$_2$/WSe$_2$ heterobilayer with a twist angle of $\theta$ - 58°. Scale bar: 5 μm. **b** Schematic of a twisted heterobilayer exhibiting a moiré pattern. It can host different excitonic states, such as localized IXs (red), delocalized IXs (black), and biexcitons (blue). **c** Band diagram of the heterobilayer showing the two typical exciton transitions: spin-singlet IX (S) and spin-triplet IX (T). **d** Semilog plot of the PL spectrum of the heterobilayer at 4.9 K in the intermediate excitation regime with S and T emissions marked.

## Trapped interlayer excitons

We now turn to the PL characteristics of the heterobilayer in the weak excitation regime for investigating the localization of IXs. To this end, we use a pump power of 33 nW and record a PL map of the sample at 4.9 K (Fig. 2a). We observe sharp emission lines with linewidths of 100 µeV in the PL spectrum (right panel of Fig. 2a), which is consistent with previous work[23] (see Supplementary Fig. S3 for additional spectra). We denote this emission feature as type-i and the related IXs as type-i IXs. The type-i IXs show a clear saturation behavior with a low saturation power of 0.91 µW (inset of Fig. 2a), which is similar to previously reported single-photon emitters in MoSe$_2$/WSe$_2$ heterobilayers[24]. The observed type-i IXs show co-polarized PL emission with the circularly polarized pump (see Fig. 2b), manifesting that the trapped IXs are of T-type in our H-stacked MoSe$_2$/WSe$_2$ heterobilayer[9,30]. Further, with increasing temperature, we observe that the PL intensity of the type-i IXs decreases (Fig. 2d). By fitting the temperature-dependent PL intensity with an Arrhenius law, we extract a thermal activation energy of 4 meV (see Supplementary Note 1), which yields the depth of the type-i potential trap[26]. The type-i IX red-shifts upon temperature increase, which is due to the relatively fast PL quenching of the higher-energy states requiring less energy for delocalization.

Interestingly, at higher temperatures (Fig. 2d) we find another peak at lower energy (denoted as type-ii), which dominates the PL spectrum (see Supplementary Fig. S4). This temperature-dependent behavior can be explained by the presence of deeper potential traps that host type-ii IXs. To further explore the nature of the type-ii IXs, the sample temperature is raised to about 40 K to suppress type-i IXs. Then, the PL emission of type-ii IXs features multiple broad emission peaks in the intermediate excitation regime (at an excitation power of 13.2 µW, see Fig. 3a), similar PL spectra measured at other positions on the heterobilayer are shown in Supplementary Fig. S5. Figure 3a shows the evolution of these multiple peaks with pump power. We find that the lowest energy state at 1.342 eV dominates the PL spectrum at the lowest pump power, while the PL intensity of higher-energy states lying between 1.352 eV and 1.37 eV grows faster than that of the lower-energy

states with increasing pump power. By integrating the PL intensity of those peaks, we extract a clear sublinear power law behavior (Fig. 3b), which indicates that the lower-energy states saturate faster than higher-energy states. Note that the delocalized T-IX shows a linear, power-dependent behavior instead. This exciton-filling feature reflects the presence of quantized energy levels originating from the moiré potential, in agreement with previous works about moiré IXs[8,14,31]. To estimate the depth of the moiré potential traps, we record the integrated PL intensity of type-ii IXs as a function of temperature and fit it with an Arrhenius equation. We obtain a value of 27 meV (see Supplementary Fig. S6), which is significantly larger than the reported defect potential in monolayer WSe$_2$[32] or MoSe$_2$[33] (<10 meV). This moiré potential of finite depth will result in a decreasing energy spacing between quantized levels within the potential and, eventually, converge into a continuum, which is illustrated in Fig. 3c. Note, however, that this decreasing energy spacing can be overwhelmed by inhomogeneous broadening due to disorders in the sample[34]. The multiple peaks (with energy below 1.37 eV) originate from the radiative recombination of IX occupied in different energy levels. The absence of clear emission features between 1.37 and 1.4 eV can be understood by the small energy spacing of the adjacent levels in the moiré potential. To summarize, we can clearly identify type-ii IXs as moiré IXs.

Let us now discuss the nature of type-i IX. The widespread spatial distribution and emission energies of type-i IXs (see Supplementary Fig. S3) are in principle consistent with previous works[21,23,24,35] that relate it to moiré-trapped IXs. In that case, two physical models have been proposed: (a) the occupation of IXs in many quantized energy levels with small energy spacing defined by the moiré potential[21,35], or (b) the occupation of IXs in single quantized energy levels, whereby the broad spectral emission range is ascribed to moiré IXs trapped within moiré potential traps, which are different due to fabrication-related distortions[23]. Model (a) would predict an increased occupation possibility of IXs in high energy levels with increasing excitation. However, this exciton-filling feature has not been observed in previous works[21,35]. In our measurements, we find that with increasing pump power

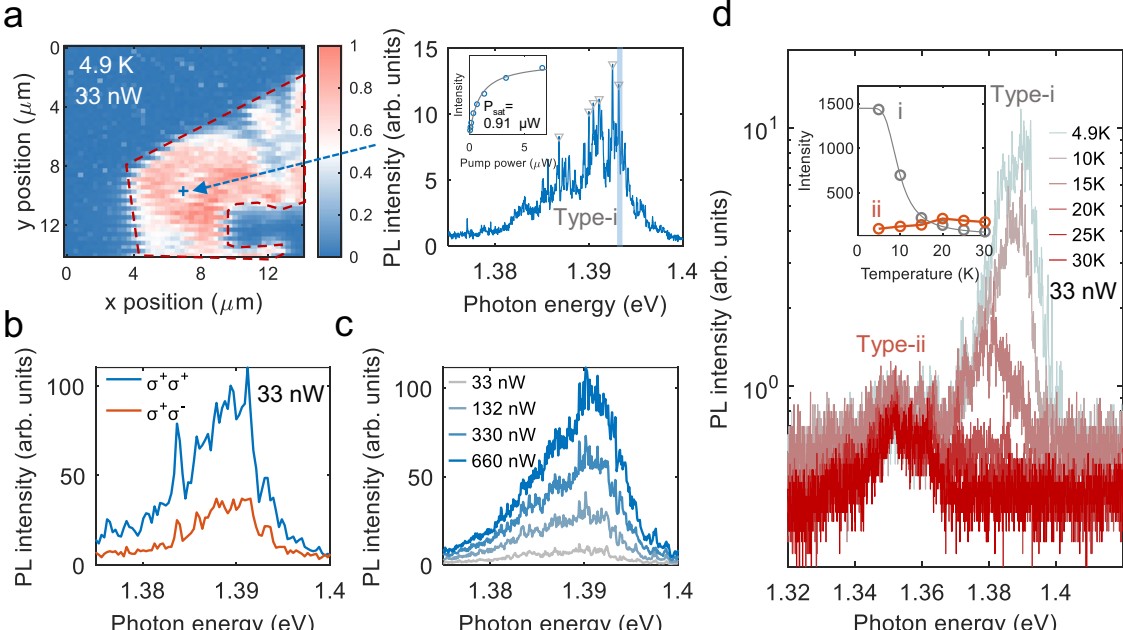

**Fig. 2 | Localized IXs (type-i IXs) in the weak excitation regime. a** PL map with a pump power of 33 nW and at 4.9 K. The right panel shows the PL spectrum of sharp emission lines (denoted as type-i emission). The integrated PL intensity of a sharp emission line in dependence on pump power is fitted with the formula $I = I_{max}/(1 + P_{sat}/P)$, where $P_{sat}$ - 0.91 µW is the saturation power. **b** Valley polarization measurement of the type-i emission. **c** Power-dependent PL spectrum. **d** PL spectrum in dependence on temperature with a pump power of 33 nW revealing two types of localized IXs. The inset shows the temperature-dependent PL intensity of two localized IX, where the integrated PL intensity of type-i IXs is fitted by an Arrhenius equation.

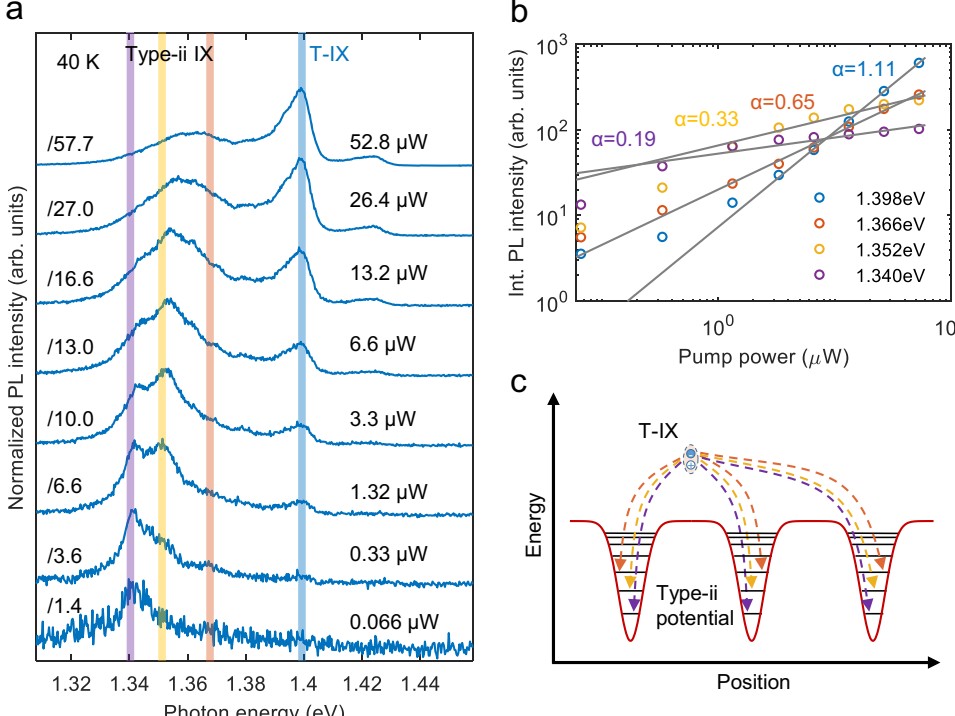

**Fig. 3 | Formation of type-ii IXs and their filling into moiré potential traps. a** PL spectrum as a function of pump power at an elevated temperature of 40 K. The PL emission of T-IXs (1.4 eV) dominates the spectrum with an increase in pump power. Each spectrum is normalized with its own peak intensity. **b** Logarithmic plot of the integrated PL intensity (with an energy range of 3 meV) of different IX states

marked in (**a**). The data are fitted by a power-law relation of the form $P^{\alpha}$. Note that the deviation from a linear dependence on the integrated PL intensity for the T-IX at pump powers below 1.32 μW is due to the poor signal-to-noise ratio of the T-IX. **c** Physical picture of the moiré potential traps that confine IXs in quantized energy levels.

(Fig. 2c), the sharp emission lines emerge into broad peaks, and no clear relative intensity change between the emission peaks is observed. Hence, this overall behavior excludes model (a) and would point toward model (b), which is similar to recent works[25,36].

However, when measuring the exciton lifetime[14,23] of type-i and type-ii IXs (see Supplementary Fig. S7), we find drastically different behavior. While type-i IXs show a slow decay of 235 ns, type-ii IXs show a lifetime that is one order of magnitude shorter (23 ns, see Supplementary Fig. S7). This large difference in lifetime indicates that the two types of localized IXs originate from spatially different potential traps, which is consistent with observing the two types of emission behavior in Fig. 2d instead of a single emission feature only[37]. Theoretically, it was shown that two types of moiré-related potential traps could exist at different high symmetry registries, i.e., $H_h^h$ and $H_h^X$, having different potential depths[12]. We expect that the type-ii IX is localized at $H_h^h$, because it is predicted to host IX with relatively large oscillator strength in a sufficiently deep potential[15,38]. The emission related to these two types of traps should show opposite circular polarization behavior[12,38]. However, we find that the emission of both type-i and type-ii IXs is co-polarized with a circularly polarized pump (see Fig. 2b and Supplementary Fig. S8). Thus, type-i IXs in our sample cannot originate from the moiré potential. As mentioned above, the emission of type-i IXs is nearly suppressed at 30 K (see Fig. 2d and Supplementary Fig. S4), which is consistent with already reported defect-bound single excitons[32,39]. Additionally, even high-quality flux-grown crystals that we use for fabricating our TMD heterobilayer are expected to have at least $10^3$ point defects per 1 μm²[40]. Hence, we conclude that the type-i emission feature is related to defect-bound IX, while the type-ii feature is related to moiré-trapped IXs.

## Interlayer biexcitons

To explore the formation of many-body excitonic states with the presence of potential traps in the TMD heterobilayer, we studied the

PL emission at even higher pump powers but remaining at a low temperature of 4.9 K (see Fig. 4a). We observe three prominent resonances: peak T at 1.399 eV and peak S (both as in Fig. 1d) and peak Z at 1.424 eV, where peak T(S) is assigned to the T(S)-IX. By examining the PL intensity of the three peaks, we find that the intensity of peak Z grows faster with increasing pump power (Fig. 4a). To quantify that, we fit the integrated intensity of the three peaks with the usual power law $\propto P^{\alpha}$. The fits yield $\alpha \sim 1.41$ for the Z peak, while $\alpha \sim 0.54$ for the T peak, and $\alpha \sim 1.04$ for the S peak. The superlinear behavior of the Z peak is a typical feature of PL emission related to biexcitons. Ideally, $\alpha$ should be 2, but values between 1.2 to 1.9 have been typically observed for emission related to biexcitons in monolayer TMDs[41], TMD heterobilayers[26,42] or quantum wells[43]. We exclude other mechanisms explaining the superlinear behavior such as superradiance[44] or Bose-Einstein condensation[5], as both would come with linewidth narrowing, which we do not observe. Thus, we attribute the observed Z feature to the emission of biexcitons. Note that the measured IXX emission energy is the same as for the S-IX, which means a zero binding energy if the IXX originates from the interaction of S-IXs. Alternatively, the formation of IXXs can come from interacting T-IXs. As peak Z has a 25 meV higher energy than the T peak, the underlying exciton-exciton interaction leading to the formation of biexcitons would then be repulsive[26,42,45].

To study the formation of IXXs in our sample, we measured their PL emission in dependence on temperature. Figure 4c shows that the exponent $\alpha$ of peak Z reaches a maximum value of 1.8 at about 30 K. Interestingly, the exponent of peak T increases to a value of about 1 at the same temperature of 30 K (consistent with our measurements presented in Fig. 3) and remains at this value at higher temperatures. This threshold behavior at a temperature of 30 K observed for the Z and the T peak matches remarkably well with the temperature that quenches type-i IXs. Hence, we hypothesize that the type-i potential

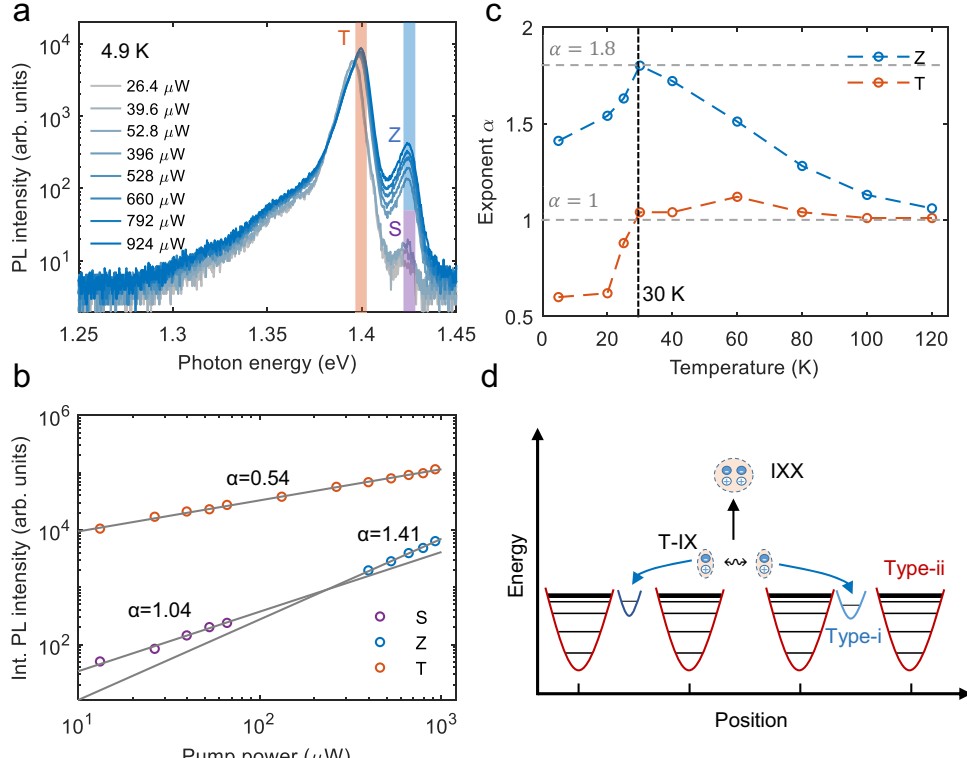

**Fig. 4 | Optical signatures of the emergence of biexcitons in the strong excitation regime. a** PL spectrum in dependence on pump power at $T$ = 4.9 K. **b** Logarithmic plot of the S, Z, and T emission intensity as a function of pump power with an integrated spectral range of 6 meV marked in (**a**). **c** Temperature-dependent slope of the integrated Z and T emission. The dashed lines are a guide to the eye. **d** Schematic illustration of the formation of the biexciton with the presence of the two types of potential traps. The blue arrows represent the formation of type-i IXs and the black arrow represents the formation of interlayer biexciton with repulsive exciton interaction.

traps have an influence on the formation of IXXs and the T-IX. This temperature-dependent change of $\alpha$ can be reproduced in another sample with a smaller twist angle (see Supplementary Fig. S9). In contrast, the moiré potential is almost unaffected at that temperature and fully filled in the strong excitation regime, permitting one to rule out its influence on the generation of IXXs.

We suggest the following physical picture (see Fig. 4d) to explain the influence of the two types of potential traps on the formation of IXXs at a low temperature of 4.9 K. With increasing pump power, the lower-energy levels of the moiré potential with larger energy spacing get gradually filled. When further increasing the pump power, the occupation of IX in higher-energy levels increases. However, the higher-energy states in the moiré potential with their small energy spacing cannot confine IXs tightly. Instead, the IX will be trapped in the type-i defect-related potential owing to repulsive dipolar interaction-driven exciton transport[46], i.e., in a high exciton density regime, the repulsive interaction will push the T-IXs out of the laser excitation spot and those excitons will be captured by the type-i potentials[46]. Then, type-i potentials can act as loss channels that reduce the IX density, where a large value is required to form IXX. Efficient quenching of the occupation of IXs in type-i potentials at a higher temperature of 30 K (see Fig. 2d) leads to the increase in the formation of IXX and, thus, to the maximum value of $\alpha$ for the Z peak. When further increasing the temperature, the value of $\alpha$ of the Z peak monotonically decreases, possibly due to temperature-induced dissociation of IXXs, resulting in an increased occupation probability of the S-state (see Supplementary Fig. S10). Future calculations are required to guide the identification of the nature of the biexciton state.

We have demonstrated the presence of two types of potential traps in the same twisted MoSe$_2$/WSe$_2$ heterobilayer, denoted as type-i and type-ii, which we identified as traps related to optically active defect states and to the moiré potential, respectively. The type-i potential is shallow and dominates the PL emission of the hetero-bilayer under weak excitation at temperatures lower than 30 K. Starting at a temperature of about 30 K, the moiré-related multi-peaked emission feature known from TMD heterobilayers becomes clearly visible at higher excitation powers. Importantly, we observe the formation of interlayer biexcitons at even higher excitation power, whose formation peaks at a temperature of 30 K. We, thus, find that the shallow type-i potential acts as a loss channel for the formation of biexcitons.

Our work clearly identifies the different temperature and excitation regimes required to observe emissions related to localized excitons and biexcitons in a MoSe$_2$/WSe$_2$ heterobilayer. Thus, our results lay the foundation for future detailed studies on the important formation dynamics of these types of excitons via ultrafast pump-probe measurements[6,47]. Understanding the different competing traps in TMD heterobilayers, foremost moiré superlattices in competition with defect-related potential traps, and the regime required to generate biexcitons is key for the use of localized excitons and biexcitons in photonic quantum technologies[12], many-body physics[18], and nonlinear optics[17].

## Methods

### Sample fabrication

TMD monolayers were prepared by mechanical exfoliation from flux-grown bulk crystals (2D semiconductors) using Nitto tape and poly-dimethylsiloxane (PDMS). The layer thickness was determined by assessing the optical contrast of microscope images of the samples and by identifying characteristics in their PL emission. We prepared MoSe$_2$ and WSe$_2$ monolayers each on a separate PDMS stamp and placed them directly on top of each other, which guarantees that the

interface between the monolayers will be clean. The heterobilayers were aligned by the straight edges of the TMD monolayers and stacked with the PDMS method[48]. The twist angles are determined by the straight edges of the monolayers and the H-type stacking is confirmed by polarization-resolved second harmonic generation (SHG) measurements (see Supplementary Fig. S1). Thin-hBN flakes used for encapsulating the heterobilayer were prepared on $SiO_2/Si$ substrate. The TMD heterobilayer and the hBN flakes were finally picked up by polycarbonate (PC) film and then placed onto fused silica[49]. The remaining PC film was removed by 1165 remover, followed by acetone and isopropyl alcohol solution. The high-quality samples we present in our work are characterized by almost bubble-free TMD heterobilayers when examined with an optical microscope and at the same time show good interlayer coupling evaluated from PL measurements at room temperature. Despite assembling many TMD heterobilayers, we only considered such high-quality heterobilayers for hBN encapsulation as used for this work.

## Optical spectroscopy

The schematic of the measurement setup for PL measurements is shown in Supplementary Fig. S11. For PL measurements, a 1.7 eV CW laser diode (LP730-SF15, Thorlabs) was used to pump the samples placed in a closed-cycle cryostat (AttoDry 800, Attocube) and a low-temperature objective with a numerical aperture (NA) of 0.81 (LT-APO/NIR/0.81, Attocube) was used for both excitation and collection. The laser light was blocked through long-pass spectral filters, and the PL signals were sent to a grating spectrometer (SR500i, Andor) with a cooled silicon array camera (DU416A-LDC-DD, Andor). Hyperspectral imaging is performed by moving the sample position with a scanner (ANSxy100lr/LT, Attocube) while taking the PL spectrum with the spectrometer. The data of the right panel of Fig. 2a are acquired with a grating of 1200 grooves/mm, i.e., with a high spectral resolution of 35 µeV. The remaining data in Fig. 2 are acquired with a grating of 600 grooves/mm (a resolution of 280 µeV), which is a good trade-off between spectral resolution and sensitivity. The data in Figs. 1, 3, and 4 are acquired with a grating of 150 grooves/mm (a resolution of 800 µeV). Polarization-resolved PL measurements are performed by using a quarter-wave plate (QWP) placed after a beamsplitter (BS). Linear polarized laser emission is converted into circularly polarized light through the QWP to excite the sample (see Supplementary Fig. S11 for details).

Time-resolved PL measurements were performed using a time-correlated single-photon counting technique with a time tagger (quTAG, qutools). We excited samples with a 1.759 eV pulsed laser (LDH-IB-705-B, PicoQuant) with a tunable repetition rate. The PL signals with energies lower than 1.71 eV were sent to a single-photon detector (SPCM-AQRH-15-FC, Excelitas).

Polarization-dependent SHG measurements were used to identify the type of stacking of our heterobilayers and were carried out with an excitation energy of 1.292 eV (repetition rate 2 kHz) from an amplified Ti:sapphire femtosecond laser system (Spectra-Physics Solstice Ace). The polarization orientation of the excitation beam was tailored by rotating an half-wave plate (HWP). The laser light after the HWP was focused onto the sample by a 40x objective lens (NA = 0.75, Nikon). The transmitted SHG signal was collected by another 40x objective lens (NA = 0.5, Nikon) and passed through a linear polarizer. A 700-nm short-pass filter was placed after the polarizer to cut off the excitation beam. The final signal was detected by a photomultiplier tube (PMT) (Hamamatsu).

## Data availability

The data generated in this study and supporting the manuscript figures including those in the Supplementary Information have been deposited in the Zenodo database under accession code https://doi.org/10.5281/zenodo.10018495[50].

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

## Acknowledgements

H.F. and W.W. acknowledge support by Olle Engkvists Stiftelse, Carl Tryggers Stiftelse, and together with S.D. by Chalmers Area of Advance Nano. W.W. acknowledges support from the Knut and Alice Wallenberg Foundation through a Wallenberg Academy Fellowship. Samples were fabricated in the Myfab Nanofabrication Laboratory at Chalmers. S.D. acknowledges financial support from European Union Graphene Flagship (Core 3, No. 881603), 2D TECH VINNOVA center (No. 2019-00068). Q.L. and S.X. acknowledge the support from the Independent Research Fund Denmark (project no. 9041-00333B and 2032-00351B), Direktør Ib Henriksens Fond, and Brødrene Hartmanns Fond. Y.Z. and Z.S. acknowledge the support from Horizon Europe (HORIZON) Project: ChirLog (101067269), the Academy of Finland (grants 314810, 333982, 336144, 336818, 352780, and 353364), Academy of Finland Flagship Programme (320167, PREIN), the EU H2020-MSCA-RISE-872049 (IPN-Bio), and ERC advanced grant (834742). E.M. acknowledges support from Deutsche Forschungsgemeinschaft (DFG) via CRC 1083 (project B09).

## Author contributions

H.F., S.D., and W.W. conceived the project. H.F. designed and fabricated the devices. H.F. and Q.L. performed PL characterizations. H.F., Q.L., S.D., and W.W. analyzed the data. Y.Z. and Z.S. performed the polarization-resolved SHG measurements. E.M., J.T., and S.X. provided support in moiré physics. All authors contributed to the discussion and writing of the manuscript.

## Funding

## Competing interests

The authors declare no competing interests.
