## [Peer Review File · Nature Communications]

Reviewers' Comments:

Reviewer #1:

Remarks to the Author:

The paper aims at exploring the relation between three kinds of interlayer excitons (IXs) (i.e., localized IXs, delocalized IXs, and biexcitons) in the H-stacked MoSe₂/WSe₂ heterobilayers. The authors fabricated the two H-stacked MoSe₂/WSe₂ heterobilayers with twist angle of near 58.2° and 59.59°. They then performed PL measurements with pumping the samples and obtained different PL spectrums by changing pump powers and temperature. The two typical IX emission features and two types of potential traps in the twisted MoSe₂/WSe₂ heterobilayers are investigated, and the formation of interlayer biexcitons is observed. Based on their studies they conclude that "the different temperature and excitation regimes required to observe emissions related to localized excitons and biexcitons in a MoSe₂/WSe₂ heterobilayers are identified clearly".

I think this work is certainly of great interest to the community and of potential high impact, and I support the publication of this paper on Nature Communications after addressing several minor questions.

1. Could you describe further the SHG method for measuring angle? There is an error of ± 0.5 degrees by using this measurement as mentioned in your paper. How can you further prove that there is no obvious atomic reconfiguration in your sample? And is there a periodic moiré supercell in the H-stacked MoSe₂/WSe₂ heterobilayers with a twist angle of near 58.2°? And what is the size of the moiré supercell?
2. Does the energy required for electron transitions to singlet and triplet states change with interlayer translation between MoSe₂ and WSe₂ monolayers, and how does this change trend? The authors could investigate the changing trend by using the first-principles calculation.
3. You mentioned in the article, "Theoretically, it was shown that two types of moiré-related potential traps can exist at different high symmetry registries, i.e., H_h^h and H_h^X, having different potential depths", Then which position does the depth ~ 27 meV for type-ii moiré potential measured by the Arrhenius equation correspond to your sample? Or the average potential depth over the entire sample? The authors should point out this in this paper.
4. As seen from Fig. S10, peak Z has a slightly higher energy than peak S, whereas, in Fig. 4a, the measured IX emission energy is the same as for the S-IX. The authors should further discuss this difference and give more possible explanations for the origin of the formation of a biexciton in the H-stacked MoSe₂/WSe₂ heterobilayers. An insightful discussion on that will be quite helpful.

Reviewer #2:

Remarks to the Author:

The authors study the photoluminescence response of marginally twisted MoSe₂/WSe₂ heterobilayers over a range of laser excitation powers and temperatures. In particular, they study the response in the energy range that usually corresponds to interlayer excitons (IXs). At low temperature and intermediate excitation power, they measure the triplet and singlet interlayer excitons reported in extant literature. They measure two types of trapped excitons: "type-i" excitons, which are attributed to excitons trapped in shallow defects, are associated with narrow linewidth peaks measured at low excitation powers and low temperatures, and "type-ii" excitons, which are attributed to excitons trapped in the moiré potential, are associated with wider linewidth peaks whose separation distance is dependent on twist angle, measured at intermediate excitation power. At strong excitation powers, they measure a signal they attribute to interlayer biexcitons.

The types of features associated with the singlet and triplet delocalized excitons and the trapped type-i and type-ii excitons have been measured before in heterobilayers, but the value of this work is the systematic study of dependence on pump power and temperature in each of the regimes of interest, as well as measurements of the circular polarization of the signals of interest. The authors use these studies to identify the origins of the signals in the various regimes and to tell an

overarching story of the evolution of interlayer exciton behavior in a heterobilayer as a function of pump power, including the role played by two types of potential traps (defect traps for type-i and moire potential trap for type-ii).

This kind of careful experimental study is very valuable. This is especially the case for twisted heterobilayers, since it is extremely—often prohibitively—computationally expensive to model these systems from first principles due to the large periodic length and due to the fact that accurate treatment of excitons requires treatment of interactions (GW level of theory) and solving the Bethe-Salpeter equation (BSE), both additionally computationally expensive. I find the authors' manuscript to be clear, well organized, and well written. I find their explanation fairly convincing, but there are just a few points that I think require clarification, which I enumerate below.

- 1) The authors state that "type-i IXs show co-polarized PL emission...manifesting that the trapped IXs are of T-type." The authors identify Ref. 23 as a report of these same type of type-i IXs. However, in Ref. 23, the polarization of the IX signal depends on the twist angle of the sample (co-polarized for 57 and 20 degrees and cross-polarized for 2 degrees). Do the authors expect twist angle to effect whether the type-i excitons are of T or S type?
- 2) In their argument that the type-i and type-ii IXs do not originate from IXs trapped in different regions of the moire potential, the authors cite ref. 12, which asserts that excitons localized in the R^h_h and R^X_h regions will have opposite circular polarizations, while at R^M_h "light coupling is forbidden." Have the authors checked that the phase behavior is the same (or similar) for the H-stacked (near 60-degree) bilayers that they are studying?
- 3) In the discussion of bilayer excitons, the authors cite Refs. 38 and 39 as evidence that biexcitons in monolayer TMDs have been observed with α between 1.2 and 1.9. (The authors' $\alpha=1.41$.) However, in refs. 38 and 39, the Intensity goes like (single exciton intensity) $^\alpha$, whereas in the authors' manuscript, the intensity goes like (pump power) $^\alpha$. If the biexcitons are formed from singlet excitons, whose intensity scales linearly with pump power, this doesn't affect the authors' argument. But if the biexcitons are formed from triplet excitons, I think the intensity scales like (triplet exciton intensity) $^\alpha$, with α greater than 2. Does this affect the authors' conclusions about which delocalized IXs bind to form the biexcitons? If it's the singlet IXs that form the biexcitons, can the authors provide a reference for previous observations of biexcitons with zero binding energy?
- 4) Can the authors speculate about why the spacing between peaks originating from the type-ii excitons (Fig. 3a) does not decrease with increasing photon energy, as might be expected from the decreasing space between energy levels in the moire traps?
- 5) The authors estimate the depth of the potential traps by fitting temperature dependent data with the Arrhenius equation. Can the authors comment on what the "a" value is in these fits and whether it makes sense for this material?
- 6) Just as a minor clarity issue: I think it would be helpful to include the laser power on panels a (right) and d in Figure 2. It would also be helpful to clarify that all of the sharp peaks in the right part of panel a correspond to the type-i IXs, and the highlighted peak is simply the peak for which the power dependence is studied.

Reviewer #3:

Remarks to the Author:

General Comments: The primary take away for this paper is that the authors have identified two different types of potential traps that interlayer excitons fall into depending on the power and temperature of the system. They also identified a biexciton peak in the high-power regime that arises from interacting spin-singlet interlayer excitons which the type-i potential trap acts as a loss channel. I liked how thorough the paper was, there were lots of additional supplemental data plus relevant sources that collaborated stated assumptions and conclusions. I disliked how confusing some parts of the paper were which could be improved upon.

Major Changes:

1. The outline/flow of the section "Trapped interlayer excitons" needs some work. The discussion for the type-i IX is not discussed until the very end of the section which is very odd and does not flow well in the paper. The discussion for these IX are not dependent on the data shown in figure 3

but rely on some of the supplementary data and figure 2 so it would be more natural to talk about where you think these potential traps are coming from for type-i before talking about type-ii. If you want to talk about the assignment of where the potential traps are coming from for both types at the same time, i.e. the end of this section, then I would change the way that this is done. Talk about all the relevant data including the lifetime data and then paint a clear picture of how these traps are different and where you think they are arising from.

2. You should add some word labels to figure 4d. It is clear what this is trying to illustrate to someone who reads the entire paper, but some people only look at the figures. Also figures 2e, 3c, and 4d are very similar and 2e is not even talked about in the paper. So, I am wondering if all three of these need to be included, or if you can combine 2e and 4d for when you talk about the potential trap origins at the end of the section.

3. Figure 3b, the power log plot. The blue data seems like two linear lines indicating two different power regimes instead of only one and the linear line seems to be only following the high-power points and not the low-power points, was this on purpose? For the 0.066uW power, some of the peaks are completely gone so how do you have a value for this power for that peak? If you did not include the 0.066uW data, what power range did you include? I count 8 points on the graph, but only six powers for figure 3a.

Minor Changes:

1. Figure 1a, some of the text is hard to read, some maybe try some other colors. The red-dash line is also not connected, fully outlining the HS section.

2. Figure 4c, maybe try to make it more obvious that this graph's y-axis is alpha like by adding a line where $\alpha = 1$ and $\alpha = 1.8$ is.

3. Supplemental 1 says "twist angle is measured 10 times". What does this mean? Is this done by the same person? If the angle is estimated by examining the optical pictures, then just say so. The edges of each TMD are fairly obvious, so I am not sure what you mean by it was measured 10 times. That sounds like you used a program or instrument to measure the twist angle, but if it was done by optical inspection then that is an estimation even if it is done 10 times. Especially if it is done by the same person because they can just draw the same lines each time. Or did you use ten different pictures that the twist angle was estimated from? The SHG data makes it clear that the device is H-stacked instead of R-stacked, so the precise angle does not seem relevant here and getting a precise angle by human eyes from a single picture is hard.

General Questions:

1. How many devices of this structure were measured with collaborating data? The supplemental data shows one other device, but that device does not have the type-ii PL emission.

2. How did you deal with the PDMS residue for fabrication?

3. Do you have polarization data for the type-i localized IXs similar to figure S8? I know that the polarization is shown for the T and S peaks, but maybe also add the polarization for the low power type-i peaks to show that they are indeed also co-polarized.

[revised manuscript text omitted]

how thin?
→ Prob. dielectric

why?
What about the
one in the supplemental
@ $\theta = 59.59^\circ$?
→ no type-i emission
was seen 59

We first pump the heterobilayer in an intermediate excitation regime
(pump power of $66 \mu\text{W}$) by a continuous wave (cw) $\sim 730 \text{ nm}$ (corresponding
to $\sim 1.7 \text{ eV}$) laser near-resonant to the intralayer exciton of WSe₂. Two inter-
layer excitonic states (peak at $\sim 1.4 \text{ eV}$ and $\sim 1.425 \text{ eV}$, labeled as T and S,
respectively) are observed in the PL emission of the heterobilayer (Fig. 1d).
The two peaks are further characterized by polarization-dependent PL mea-
surements, whereby the S-IX shows the expected cross-polarized and the
T-IX co-polarized emission with the pump (see Fig. S2). The observed energy
splitting of $\sim 25 \text{ meV}$ together with the measured valley polarization of the
two peaks are in agreement with previous measurements of the SOC-induced
singlet and triplet interlayer excitons [9, 30]. - excellent

why the
use of this
verb: could
use quite

Trapped interlayer excitons

We now turn to the PL characteristics of the heterobilayer in the weak exci-
tation regime for investigating the localization of IXs. To this end, we use a
pump power of 33 nW and record a PL map of the sample at 4.9 K (Fig. 2a).
We observe sharp emission lines with linewidths of $\sim 100 \mu\text{eV}$ in the PL spec-
trum (right panel of Fig. 2a), which is consistent with previous work [23] (see
Fig. S3 for additional spectra). We denote this emission feature as type-i and
the related IXs as type-i IXs. The type-i IXs show a clear saturation behav-
ior with a low saturation power of $\sim 0.91 \mu\text{W}$ (inset of Fig. 2a), which is
similar to previously reported single-photon emitters in MoSe₂/WSe₂ hetero-
bilayers [24]. The observed type-i IXs show co-polarized PL emission with the
circularly polarized pump (see Fig. 2b), manifesting that the trapped IXs are
of T-type [9, 30]. Further, with increased temperature, we observe that the
PL intensity of the type-i IXs decreases (Fig. 2d). By fitting the temperature-
dependent PL intensity with an Arrhenius law, we extract a thermal activation
energy of $\sim 4 \text{ meV}$ (see Supplementary Note 1), which yields the depth of the
type-i potential trap [26]. The type-i IX red-shifts upon temperature increase.

Fig. 2 Localized IXs (type-i IXs) in the weak excitation regime. (a) PL map with a pump power of 33 nW and at 4.9K. The right panel shows the PL spectrum of sharp emission lines (denoted as type-i emission). The integrated PL intensity of a sharp emission line with pump power is fitted with the formula: $I = I_{max}/(1 + P_{sat}/P)$, where $P_{sat} \sim 0.91 \mu W$ is the saturation power. (b) Valley polarization measurement of the type-i emission. (c) Power-dependent PL spectrum. (d) PL spectrum in dependence on temperature with a pump power of 33 nW revealing two types of localized IXs. The inset shows the temperature-dependent PL intensity of two localized IX, where the integrated PL intensity of type-i IXs is fitted by an Arrhenius equation. (e) Physical picture of type-i potential traps that confine the T-IXs. A potential depth of ~ 4 meV is derived from the thermal activation energy.

which is due to the relatively fast PL quenching of the higher energy states requiring less energy for delocalization.

The widespread spatial distribution and emission energies of type-i IXs (see Fig. S3) are in principle consistent with previous works [21, 23, 24, 31] that relate it to moiré-trapped IXs. In that case, two physical models have been proposed: (a) the occupation of IXs in many quantized energy levels with small energy spacing defined by the moiré potential [21, 31], or (b) the occupation of IXs in single quantized energy levels, whereby the broad spectral emission range is ascribed to moiré IXs trapped within moiré potential traps, which are different due to fabrication-related distortions [23]. Model (a) would predict an increased occupation possibility of IXs in high energy levels with increasing excitation. However, this exciton-filling feature has not been

Can you still see it at low T? \rightarrow S4
 Are you sure this is not bkgr? \rightarrow S4
 how many devices of this were measured w/ collaborating data

observed in previous works [21, 31]. In our measurements, we find that with
 increasing pump power (Fig. 2c), the sharp emission lines emerge into broad
 peaks, and no clear relative intensity change between the emission peaks is
 observed. Hence, this overall behavior excludes model (a) and would point
 toward model (b), which is similar to recent works [25, 32].

Did you measure at higher T than 40K?
 56 3.8g4c

look all equally spaced, and 3 of the 4 do not fit the data

~~look all equally spaced, and 3 of the 4 do not fit the data~~

so T-IX is the 2. same as Type-II traps
 T-IX fall into diff. traps
 @ 5K type-II
 @ 40K type-ii

very low intensity

**Fig. 3** Formation of type-II IXs and their filling into moiré potential traps. (a) PL spectrum
 as a function of pump power at an elevated temperature of 40K. The PL emission of T-
 IXs (~1.4eV) dominates the spectrum with an increase in pump power. Each spectrum is
 normalized with its own peak intensity. (b) Logarithmic plot of the integrated PL intensity
 (with an energy range of ~3 meV) of different IX states marked in (a). The data are fitted
 by a power-law relation of the form P^α . (c) Physical picture of the moiré potential traps
 that confine IXs in quantized energy levels.

We will come back to the origin of the type-I emission feature, but will first
 discuss the spectral feature that we observe at higher temperatures (Fig. 2d),
 where we find that another peak with lower energy (denoted as type-II) dom-
 inates the PL spectrum (Fig. S4). This temperature-dependent behavior can
 be explained by the presence of deeper potential traps that host type-II IXs.
 To further explore the nature of the type-II IXs, the sample temperature is
 raised to about 40K to suppress type-I IXs. Then, the PL emission of type-
 ii IXs features multiple broad emission peaks in the intermediate excitation
 regime (at an excitation power of 13.2 μW, see Fig. 3a), similar PL spectra
 measured at other positions on the heterobilayer are shown in Fig. S5. Fig. 3a

put this sooner, then describe what's happening

shows the evolution of these multiple peaks with pump power. We find that the lowest energy state at ~ 1.342 eV dominates the PL spectrum at the lowest pump power, while the PL intensity of higher-energy states lying between 1.352 eV and 1.37 eV grows faster than that of the lower-energy states with increasing pump power. By integrating the PL intensity of those peaks, we extract a clear sublinear power law behavior (Fig. 3b), which indicates that the lower energy states saturate faster than higher energy states. Note that the delocalized T-IX shows instead a linear, power-dependent behavior. This exciton-filling feature reflects the presence of quantized energy levels originating from the moiré potential, in agreement with previous works about moiré IXs [8, 14, 33]. To estimate the depth of the moiré potential traps, we record the integrated PL intensity of type-ii IXs as a function of temperature and fit it with an Arrhenius equation, which yields a value of ~ 27 meV (see Fig. S6). This moiré potential of finite depth will result in a decreasing energy spacing between quantized levels within the potential and, eventually, converge into a continuum, which is illustrated in Fig. 3c. The multiple peaks (with energy below 1.38 eV) originate from the radiative recombination of IX occupied in different energy levels. The absence of clear emission features between 1.37 eV and 1.4 eV can be understood by the small energy spacing of the adjacent levels in the moiré potential. Hence, we identify type-ii IXs as moiré IXs.

Let us now come back to examine more closely the nature of type-i IX. To this end, we also measured the exciton lifetime [14, 23] of type-i and type-ii IXs (see Fig. S7). Compared to type-i IXs with a slow decay of ~ 235 ns, the type-ii IXs show a lifetime that is one order of magnitude shorter (~ 23 ns, see Fig. S7). This large difference in lifetime indicates that the two types of localized IXs originate from spatially different potential traps, which is consistent with observing the two types of emission behavior in Fig. 2(d), instead of a single emission feature only [34]. Theoretically, it was shown that two types of moiré-related potential traps can exist at different high symmetry registries, i.e., H_h^h and H_h^X , having different potential depths [12]. It is predicted that the emission related to these two types of traps should show opposite circular polarization behavior [12]. However, we find that the emission of both type-i and type-ii IXs is co-polarized with a circularly-polarized pump (see Fig. 2b and Fig. S8). Thus, type-i IXs in our sample cannot originate from the moiré potential. As mentioned above, the emission of type-i IXs is nearly suppressed at 30 K (see Fig. 2d and Fig. S4), which is consistent with already reported defect-bound single excitons [35, 36]. Additionally, even high quality flux-grown crystals that we use for fabricating our TMD heterobilayer are expected to have at least 10^3 point defects per $1 \mu\text{m}^2$ [37]. Hence, we conclude that the type-i emission feature is related to defect-bound IX.

Interlayer biexcitons

To explore the formation of many-body excitonic states with the presence of potential traps in the TMD heterobilayer, we studied the PL emission at even

$\alpha < 1$
sublinear
vs linear
 $\alpha \approx 1$

more moiré
traps? not defect

how do you
know this is
not defect

@ 40K??

yes S6
~~yes, the paper~~
persists up
to 80K

then how
do you know
one is not
from a vacancy?

one is from
defect

range of powers
IX & X
mix & more obvious

[revised manuscript text omitted]
 that support the findings of this study are available from the corresponding authors upon reasonable request.

Code availability. The code used in this study is available from the corresponding author upon reasonable request.

Acknowledgments. H. F. and W. W. acknowledge support by Olle Engkvists Stiftelse, Carl Tryggers Stiftelse, and together with S. D. by Chalmers Area of Advance Nano. W. W. acknowledges support by the Knut and Alice Wallenberg Foundation through a Wallenberg Academy Fellowship. Samples were fabricated in the Myfab Nanofabrication Laboratory at Chalmers. S. D. acknowledges financial support from European Union Graphene Flagship (Core 3, No. 881603), 2D TECH VINNOVA center (No. 2019-00068). Q. L. and S. X. acknowledge the support from the Independent Research Fund Denmark (project no. 9041-00333B and 2032-00351B), Direktor Ib Henriksens Fond, and Brødrene Hartmanns Fond. Y. Z. and Z. S. acknowledge the support from Horizon Europe (HORIZON) Project: ChirLog (101067269), the Academy of Finland (grants 314810, 333982, 336144, 336818, 352780, and 353364), Academy of Finland Flagship Programme (320167, PREIN), the EU H2020-MSCA-RISE-872049 (IPN-Bio), and ERC advanced grant (834742). E. M. acknowledges support from Deutsche Forschungsgemeinschaft (DFG) via CRC 1083 (project B09).

Author contributions. H. F., S. D., and W. W. conceived the project. H. F. designed and fabricated the devices. H. F. and Q. L. performed PL characterizations. H. F., Q. L., S. D., and W. W. analyzed the data. Y. Z. and

507 Z. S. performed the polarization-resolved SHG measurements. E. M., J. T.,
and S. X. provided support in moiré physics. All authors contributed to the
discussion and writing of the manuscript.

**Competing interests.** The authors declare no competing interests.

Supplementary Text

Supplementary Note 1. Estimation of the depth of potential traps

The temperature-dependent PL intensity change can be described by the Arrhenius equation $I = I_0[1 + a \exp(-E_A/k_B T)]^{-1}$ [46], where E_A is the thermal activation energy and a is a coefficient that is related to the quantum yield of the host material [47]. Through fitting, we obtain an E_A of ~ 4 meV for the shallow type-i potential traps and E_A of ~ 27 meV for the type-ii moiré potential.

Supplementary Figures

Fig. S1 Characterization of twist angle. (a) Optical image of the stacked MoSe₂/WSe₂ heterobilayer on PDMS. The twist angle is measured 10 times according to the straight edges and determined to be $58.2^\circ \pm 0.5^\circ$. Scale bar: 10 μm . (b) Polarization-resolved SHG measurement. The SHG intensity of the heterobilayer is remarkably weak compared to monolayer MoSe₂, confirming that the stack type is H-stack [48].

by the same person? measured?

show all four for completeness?

Fig. S2 Valley polarization of S and T with a pump power of $66 \mu\text{W}$ at 4.9 K.

conclude is from defect bound excitons
each spot looks a little different

Fig. S3 Spatial distribution of type-i IXs. PL map obtained by integrating the PL spectrum between ~ 1.333 and ~ 1.425 eV at 4.9 K. The sharp emission lines (denoted as type-i) from the marked positions are shown in the right panel.

Does the type-ii @ 40K look wildly different @ different spots?
no -> 55

Fig. S4 PL emission as a function of temperature with a pump power of 33 nW. Each spectrum is normalized with its own peak intensity.

*still slightly different,
but not as wildly diff as
the type-i*

**Fig. S5** PL map of type-ii localized IXs with a pump power of 330 nW at 40 K. The multi-
peak feature can be observed from various positions, however, the energy spacing between
adjacent peaks varies with position. Such a variation will make it difficult to observe a twist
angle-dependent energy spacing when comparing different samples.

**Fig. S6** Estimation of the depth of the type-ii potential via fitting by the Arrhenius
equation. Note that the data is acquired from a different position than the inset shown in
Fig. 2d of the main text.

Fig. S7 Time-resolved PL dynamics of type-i and type-ii IXs. The solid lines represent the biexponential fits to the data, yielding the lifetime of type-i (~ 40 ns and ~ 235 ns) and type-ii (~ 3 ns and ~ 23 ns) IX. The lifetime of type-i (ii) localized IXs is measured at 4.9 K (40 K). Note that the measured type-i and type-ii IX lifetime is similar to the reported lifetime of sharp emission lines [23] and moiré-related peaks [14], respectively.

Fig. S8 Valley polarization of type-ii localized IXs with a pump power of 330 nW.

**Fig. S9** Reproducibility of the temperature-dependent PL emission behavior in another
sample. (a) optical image of the hBN-encapsulated MoSe₂/WSe₂ heterobilayer. (b) Power-
dependent PL spectrum in the high excitation regime. (c) Temperature-dependent change of
exponent α . (d) Evolution of PL spectrum with temperature. Each spectrum is normalized
with its own peak intensity. No clear type-II emissions emerge with increasing temperature,
which is possibly due to the atomic reconstruction for the small twisted ($< 1^\circ$) heterobi-
layer [28].

was this the
only other
hs measure?

**Fig. S10** Evolution of PL spectrum with a pump power of 924 μW. With increasing tem-
perature to 80 K, the IXX gradually vanishes (see Fig. 4c) and the S-IX takes over the
emission peak. The increasing (decreasing) PL intensity of S(T)-IX with temperature is con-
sistent with previous work [9].

Fig. S11 Schematics of the PL measurement setup. LP: linear polarizer. HWP: half-waveplate. BS: beam splitter. SMF: single-mode fiber. The PL signal is sent to the spectrometer for measuring the PL spectrum, to the monochromator to filter the desired emission peak and measure exciton lifetime. LP1 is set to be horizontally linear polarized. LP2 and QWP are only used for valley polarization measurements. QWP is used to convert the linearly polarized laser emission to circularly polarized and then excite the sample. The circularly-polarized PL signal will be converted into a linear polarized signal, which is analyzed by LP through rotating HWP2. QWP and LP2 are removed when performing power-

***** Conventions *****

We answer the referees' criticism point-by-point. To this end, we keep the order of the referees' comments and mark it > **in blue and bold font** <. Our answers are in black and normal text. For each point, we indicate which parts of the manuscript have been changed \gg in black in response to the referee's comments.

When we refer to Figures, Tables, Citations etc., we use the numbering of the revised version of the manuscript and supplemental material.

We add a marked-up version of our revised manuscript, where **added text is in blue, normal text, and removed text is in red, strikethrough text.**

***** Addressing Report of Reviewer 1 *****

We thank Reviewer 1 for the time and effort employed in the review and their comments. We address all comments in the following point-by-point.

> **The paper aims at exploring the relation between three kinds of inter-layer excitons (IXs) (i.e., localized IXs, delocalized IXs, and biexcitons) in the H-stacked MoSe₂/WSe₂ heterobilayers. The authors fabricated the two H-stacked MoSe₂/WSe₂ heterobilayers with twist angle of near 58.2 and 59.59. They then performed PL measurements with pumping the samples and obtained different PL spectrums by changing pump powers and temperature. The two typical IX emission features and two types of potential traps in the twisted MoSe₂/WSe₂ heterobilayers are investigated, and the formation of interlayer biexcitons is observed. Based on their studies they conclude that "the different temperature and excitation regimes required to observe emissions related to localized excitons and biexcitons in a MoSe₂/WSe₂ heterobilayers are identified clearly".**

I think this work is certainly of great interest to the community and of potential high impact, and I support the publication of this paper on Nature Communications after addressing several minor questions. <

We thank the referee for the clear summary of our work and for acknowledging its relevance and impact for the community, and for the clear recommendation to publish our work in Nature Communications.

> **1. Could you describe further the SHG method for measuring angle? There is an error of ± 0.5 degrees by using this measurement as mentioned in your paper. How can you further prove that there is no obvious atomic reconfiguration in your sample? And is there a periodic moiré supercell in the H-stacked MoSe₂/WSe₂ heterobilayers with a twist angle of near 58.2°? And what is the**

size of the moiré supercell? <

The method we used to determine the twist angle was via looking at the straight edge of monolayers in an optical microscope. The quoted uncertainty of $\pm 0.5^\circ$ (now corrected to $\pm 0.6^\circ$, see the reply to Referee 3) is determined by the optical contrast of the edges. This method has been demonstrated (J. Wang et al., *Sci. Adv.* 5, eaax014 (2019)) as an effective way of measuring twist angles. We do not use the SHG to determine the twist angle due to the weak SHG intensity of the heterobilayer region, as a consequence of good interlayer coupling (W. Hsu, et al., *ACS Nano* 8, 2951-2958 (2014)). In this case, the uncertainty from the SHG measurement was much larger than the one from the optical microscope images.

The atomic reconstruction is usually measured via conductive atomic-force microscopy and scanning transmission electron microscopy (*Nat. Nanotechnol.* 15, 592–597 (2020)), which we do not have readily available to use. Regarding H-stacked MoSe₂/WSe₂ heterobilayers, it was demonstrated that they need to have a twist angle smaller than 1° (or larger than 59°) (V.V.Enaldiev, et al., *Phys. Rev. Lett.* 124, 206101 (2020), M. R. Rosenberger, et al., *ACS Nano* 14, 4550–4558 (2020)) to see a clear atomic reconstruction. The sample we present in the main text of our work has a twist angle of $\sim 58^\circ$, and, thus, should not be much affected by atomic reconstruction. From ref. (E. Barré, et al., *Science* 376, 406-410 (2022)), at a twist angle of 58° there is a moiré supercell predicted and also been observed in the experiment. Using ref. (Nan Zhang, et al., *Nano Lett.* 18, 7651–7657 (2018)), we compute the moiré supercell to be in our case ~ 9.4 nm.

Note that the sample we present in Fig. S9 has a twist angle of $\sim 59.6^\circ$ and, thus, should be affected by atomic reconstruction, which we now also mention in the caption of Fig. S9. This may explain why we do not observe type-ii emission (i.e., emission from moiré IX) in that sample.

» We added the following text as Supplementary Note 2: The moiré period a_M can be calculated via (N. Zhang, et al., *Nano Lett.* 18, 7651–7657 (2018)):

$$a_M = \frac{a_{\text{MoSe}_2} a_{\text{WSe}_2}}{\sqrt{a_{\text{MoSe}_2}^2 + a_{\text{WSe}_2}^2 - 2a_{\text{MoSe}_2} a_{\text{WSe}_2} \cos \theta}}$$

where a_{MoSe_2} (a_{WSe_2}) is the lattice constant of monolayer MoSe₂ (WSe₂). Considering a_{MoSe_2} (a_{WSe_2}) = 0.3288 nm (0.3280 nm) (C. Huang, et al., *Nat. Mater.* 13, 1096–1101 (2014)) and the twist angle ($\sim 58^\circ$) of the sample in the main text, we derive a moiré period of ~ 9.4 nm.

> 2. Does the energy required for electron transitions to singlet and triplet states change with interlayer translation between MoSe2 and WSe2 monolayers, and how does this change trend? The authors could investigate the changing trend by using the first-principles calculation. <

The energy spacing between S and T states is defined by the energy splitting of the MoSe₂ conduction band at the K valley (T. Wang, et al., *Nano Lett.* 20, 694–700 (2020)), which should not vary with twist angle but is determined by the difference in the exciton binding energy of T and S states on top of the spin-orbit splitting of the conduction band.

Regarding the energy of the S and T states, we would expect a redshift with decreasing twist angle according to previous work (E. Barré, et al., *Science* 376, 406-410 (2022)).

This energy shift is expected due to the hybridization of the states in a heterostructure (P. Merkl, et al., *Nat. Commun.* **11**, 2167 (2020)).

> **3. You mentioned in the article, "Theoretically, it was shown that two types of moiré-related potential traps can exist at different high symmetry registries, i.e., H_h^h and H_h^X , having different potential depths", Then which position does the depth ~ 27 meV for type-ii moiré potential measured by the Arrhenius equation correspond to your sample? Or the average potential depth over the entire sample? The authors should point out this in this paper. <**

We expect that the type-ii IX is localized at H_h^h , because it is predicted to host IX with relatively large oscillator strength in a sufficiently deep potential (X. Lu, et al., *Phys. Rev. B* **100**, 155416 (2019), E. Barré, et al., *Science* **376**, 406-410 (2022)). So far, there is no well-accepted theoretical model to predict the potential depth of H-stacked MoSe₂/WSe₂ heterobilayers.

» We have now added the following additional statement about the spatial position of type-ii IX in our manuscript: "We expect that the type-ii IX is localized at H_h^h , because it is predicted to host IX with relatively large oscillator strength in a sufficiently deep potential (X. Lu, et al., *Phys. Rev. B* **100**, 155416 (2019), E. Barré, et al., *Science* **376**, 406-410 (2022))" after our statement: "Theoretically, it was shown that two types of moiré-related potential traps can exist at different high symmetry registries, i.e., H_h^h and H_h^X , having different potential depths".

> **4. As seen from Fig. S10, peak Z has a slightly higher energy than peak S, whereas, in Fig. 4a, the measured IXX emission energy is the same as for the S-IX. The authors should further discuss this difference and give more possible explanations for the origin of the formation of a biexciton in the H-stacked MoSe₂/WSe₂ heterobilayers. An insightful discussion on that will be quite helpful. <**

Let us first clarify the difference between the measurements presented in Fig. S10 and Fig. 4a. The former shows the PL in the high excitation regime at different temperatures while the latter shows the PL at different pump powers at 4.9 K. When changing temperature, the band structure changes. In the heterobilayer system that we use in our experiments, we expect a redshift of the emission (see works T. Wang, et al., *Nano Lett.* **20**, 694–700 (2020) and L. Zhang, et al., *Phys. Rev. B* **100**, 041402 (2019)). This is also what we observe in Fig. S10.

We attribute the predominant emission at the different temperatures to the Z (high excitation regime) or S peak (low excitation regime) by performing power-dependent measurements (see Figure R1 below). We use a power law to fit the data and extract the exponent α . With increasing temperature, we gradually do not see a difference of α between low and high excitation regimes and its value is approaching 1 (also see Fig. 4c in the main text).

We expect that the formation of the biexciton is due to the repulsive exciton-exciton interaction in the high exciton density regime. The repulsive interaction will push the IXs out of the excitation spot (Z. Sun, et al., *Nat. Photon.* **16**, 79–85 (2022)) and the IXs will be captured by type-i potentials, which act as a loss channel of the T-IX and reduce the formation of the biexciton. We note that the repulsive interaction is evidenced by the

higher energy of the biexciton (Z) than that of the single exciton (T), which was discussed in the main text (lines 370-373). Therefore, the formation of the biexciton is increased when the temperature is increased to 30 K where the type-i IX is suppressed, which is evidenced by the increase of α as shown in Fig. 4c.

Figure R1: Integrated PL intensity as a function of pump power at 40 K and 80 K.

» We added the following to our supplementary material to the caption of the revised Fig. S10: "We attribute the predominant emission at the different temperatures to the Z (high excitation regime) or S peak (low excitation regime) by performing power-dependent measurements. We use a power law to fit the data and extract the exponent α , as shown in (b) and (c). With increasing temperature, we gradually do not see a difference of α between low and high excitation regimes and its value is approaching 1 (also see Fig. 4c in the main text)". We added additional discussion about the formation of biexciton in the main text, which is located in lines 391-394 of the revised manuscript.

***** Addressing Report of Reviewer 2 *****

> The authors study the photoluminescence response of marginally twisted MoSe₂/WSe₂ heterobilayers over a range of laser excitation powers and temperatures. In particular, they study the response in the energy range that usually corresponds to interlayer excitons (IXs). At low temperature and intermediate excitation power, they measure the triplet and singlet interlayer excitons reported in extant literature. They measure two types of trapped excitons: "type-i" excitons, which are attributed to excitons trapped in shallow defects, are associated with narrow linewidth peaks measured at low excitation powers and low temperatures, and "type-ii" excitons, which are attributed to excitons trapped in the moire potential, are associated with wider linewidth peaks whose separation distance is dependent on twist angle, measured at intermediate excitation power. At strong excitation powers, they measure a signal they attribute to interlayer biexcitons.

The types of features associated with the singlet and triplet delocalized excitons and the trapped type-i and type-ii excitons have been measured before

in heterobilayers, but the value of this work is the systematic study of dependence on pump power and temperature in each of the regimes of interest, as well as measurements of the circular polarization of the signals of interest. The authors use these studies to identify the origins of the signals in the various regimes and to tell an overarching story of the evolution of interlayer exciton behavior in a heterobilayer as a function of pump power, including the role played by two types of potential traps (defect traps for type-i and moire potential trap for type-ii).

This kind of careful experimental study is very valuable. This is especially the case for twisted heterobilayers, since it is extremely - often prohibitively - computationally expensive to model these systems from first principles due to the large periodic length and due to the fact that accurate treatment of excitons requires treatment of interactions (GW level of theory) and solving the Bethe-Salpeter equation (BSE), both additionally computationally expensive. I find the authors' manuscript to be clear, well organized, and well written. I find their explanation fairly convincing, but there are just a few points that I think require clarification, which I enumerate below. <

We very much acknowledge the clear summary of our work by the referee and the appreciation of the systematic measurements we present.

> 1) The authors state that "type-i IXs show co-polarized PL emission ... manifesting that the trapped IXs are of T-type." The authors identify Ref. 23 as a report of these same type of type-i IXs. However, in Ref. 23, the polarization of the IX signal depends on the twist angle of the sample (co-polarized for 57 and 20 degrees and cross-polarized for 2 degrees). Do the authors expect twist angle to effect whether the type-i excitons are of T or S type? <

First, one distinguishes R-type and H-type heterobilayers. In case of an R-type heterobilayer, the band diagram is shown in the left panel of Fig. R2. In this scenario, the ground state is S-type IX, which is spin-conserved and will dominate the emission. Therefore, one typically sees only one exciton state as in Ref. (E. Y. Paik, et al., *Nature* 576, 80–84 (2019)). In contrast, the ground state of the H-type heterobilayer (right panel of Fig. R2) is the spin-flipped T-type IX, which is brightened due to breaking of the out-of-plane mirror symmetry and shows similar oscillator strength as the S-type IX (H. Yu, et al., *2D Mater.* 5, 035021 (2018)). As a result, we observe both the S and T states, as found in previous works (T. Wang, et al., *Nano Lett.* 20, 694–700 (2020), L. Zhang, et al., *Phys. Rev. B* 100, 041402 (2019)). The different band diagrams shown in Fig. R2 explain the opposite polarization behavior of R- and H-type heterobilayers.

Figure R2: Excitonic transitions in R- and H-type MoSe₂/WSe₂ heterobilayer.

> 2) In their argument that the type-i and type-ii IXs do not originate from IXs trapped in different regions of the moire potential, the authors cite ref. 12, which asserts that excitons localized in the R_h^h and R_h^X regions will have opposite circular polarizations, while at R_h^M "light coupling is forbidden." Have the authors checked that the phase behavior is the same (or similar) for the H-stacked (near 60-degree) bilayers that they are studying? <

Ref. 12 states that "The heterobilayer moiré formed by different TMD compounds and the H-type stacking feature similar nanopatterned spin optics as the R-type MoS₂/WSe₂, whereas the potential profiles and their field dependence can have quantitative differences" with a supporting calculation shown in their Supplementary Information. More specifically, regarding H-stacked MoSe₂/WSe₂ heterobilayer, Ref. X. Lu, et al., *Phys. Rev. B* **100**, 155416 (2019) has shown that opposite circular polarization is observed at H_h^h and H_h^X regions, see Figure R3.

Figure R3: Optical oscillator strength of interlayer excitons at different sites of an H-stacked MoSe₂/WSe₂ heterobilayer (X. Lu, et al., Phys. Rev. B 100, 155416 (2019)).

» We have added ref. X. Lu, et al., Phys. Rev. B 100, 155416 (2019) to the main text.

> 3) In the discussion of bilayer excitons, the authors cite Refs. 38 and 39 as evidence that biexcitons in monolayer TMDs have been observed with alpha between 1.2 and 1.9. (The authors' alpha=1.41.) However, in refs. 38 and 39, the Intensity goes like (single exciton intensity)^α, whereas in the authors' manuscript, the intensity goes like (pump power)^α. If the biexcitons are formed from singlet excitons, whose intensity scales linearly with pump power, this doesn't affect the authors' argument. But if the biexcitons are formed from triplet excitons, I think the intensity scales like (triplet exciton intensity)^α, with alpha greater than 2. Does this affect the authors' conclusions about which delocalized IXs bind to form the biexcitons? If it's the singlet IXs that form the biexcitons, can the authors provide a reference for previous observations of biexcitons with zero binding energy? <

We note that other works use pump power as the x-axis as well, see for example (W. Li, et al., Nat. Mater. 19, 624–629 (2020) (see Figure R4), H. Zheng, et al., Light Sci. Appl. 12, 117 (2023)).

Figure R4: Identification of biexciton. (W. Li, et al., Nat. Mater. 19, 624–629 (2020)).

We agree, however, that the ideal way to obtain α is to determine the single exciton density and use that value as the x-axis. In our case, taking that PL intensity of the T-IX as the x-axis, we would then determine a value for α that is larger than 2 for the Z peak. We would attribute this unexpected over-2 behavior to the difficulty of excluding the effect from the type-i defect-related emission that is very close in energy to the T-IX emission. We believe that this close energy spacing results in a joint excitation of T-IX and type-i IX with the result that the PL of the T-IX grows sublinearly with pump power. We can

see clearly that the effect of the type-i IX on the T-IX is removed at temperatures larger than 30 K (Fig. 4c in the main text).

The binding energy of a biexciton is given by the interaction between single excitons (could be attractive or repulsive, V. I. Klimov, et al., Nature 447, 441–446 (2007)), which are required to form a biexciton. Future calculations of biexcitons in heterobilayers may then lead to a clear identification of the origin of the biexciton state.

» We added the mentioned references (V. I. Klimov, et al., Nature 447, 441–446 (2007), H. Zheng, et al., Light Sci. Appl. 12, 117 (2023), W. Li, et al., Nat. Mater. 19, 624–629 (2020)) about biexcitons in TMD heterobilayers into the main text. We added this description "Future calculations are required to guide the identification of the nature of the biexciton state" into the main text (lines 401-403) of the revised manuscript.

> **4) Can the authors speculate about why the spacing between peaks originating from the type-ii excitons (Fig. 3a) does not decrease with increasing photon energy, as might be expected from the decreasing space between energy levels in the moire traps? <**

In our measurements, the laser spot ($\sim 1 \mu\text{m}$) is far larger than the moiré period ($\sim 10 \text{ nm}$), which means that we will collect the PL signal from ~ 10000 sites. As is commonly the case (C. N. Lau, et al., Nature 602, 41–50 (2022)), the moiré period is not uniform over the sample and will, thus, lead to slightly different energies at each site. This will lead to inhomogeneous broadening of the type-ii emission, i.e., a larger linewidth of each sublevel, which makes the clear determination of energy differences harder as the moiré sublevels overlap in energy.

» We added to our revised manuscript: "Note, however, that this decreasing energy spacing can be overwhelmed by inhomogeneous broadening due to disorders in the sample (C. N. Lau, et al., Nature 602, 41–50 (2022))."

> **5) The authors estimate the depth of the potential traps by fitting temperature dependent data with the Arrhenius equation. Can the authors comment on what the "a" value is in these fits and whether it makes sense for this material? <**

The value 'a' is related to the quantum yield of a material as mentioned in our Supplementary Note 1, and can be described by the ratio of the non-radiative lifetime to the radiative lifetime. We checked the value of 'a' in previous works (H. Zheng, et al., Light Sci. Appl. 12, 117 (2023), Y. Luo, et al., 2D Mater. 6, 035017 (2019), M. Leroux, et al., J. Appl. Phys. 86, 3721–3728 (1999)) and found that these works state only the activation energy. However, in our work, the extracted value is about 160, meaning that the non-radiative lifetime is much longer than the radiative lifetime and could be explained by the strong confinement of excitons that reduces the possibility of carriers captured by non-radiative centers.

» We added this description to the suppl. note 1: "In our work, the extracted value of a is about 160, meaning that the non-radiative lifetime is much longer than the radiative lifetime. The reason could be that the strong confinement of excitons reduces the possibility of carriers captured by non-radiative centers "

> **6) Just as a minor clarity issue: I think it would be helpful to include the**

laser power on panels a (right) and d in Figure 2. It would also be helpful to clarify that all of the sharp peaks in the right part of panel a correspond to the type-i IXs, and the highlighted peak is simply the peak for which the power dependence is studied. <

We thank the referee for these suggestions that we have implemented in the revised manuscript.

» We have now added the laser power (33 nW) in the right panel of Fig. 2a and 2d and marked the type-i emission lines in the right panel of Fig. 2a. Additionally, we removed Fig. 2e according to the comments of Referee 3. The revised Figure 2 and caption are shown below:

Figure 2 Localized IXs (type-i IXs) in the weak excitation regime. (a) PL map with a pump power of 33 nW and at 4.9 K. The right panel shows the PL spectrum of sharp emission lines (denoted as type-i emission). The marked sharp emission lines are taken as an example of type-i emission and one of them is integrated to show the power-dependent PL emission behavior: $I = I_{\max}/(1 + P_{\text{sat}}/P)$, where $P_{\text{sat}} \sim 0.91 \mu\text{W}$ is the saturation power. (b) Valley polarization measurement of the type-i emission. (c) Power-dependent PL spectrum. (d) PL spectrum in dependence on temperature with a pump power of 33 nW revealing two types of localized IXs. The inset shows the temperature-dependent PL intensity of two localized IX, where the integrated PL intensity of type-i IXs is fitted by an Arrhenius equation.

***** Addressing Report of Reviewer 3 *****

> **General Comments:** The primary take away for this paper is that the authors have identified two different types of potential traps that interlayer excitons fall into depending on the power and temperature of the system. They also identified a biexciton peak in the high-power regime that arises from inter-

acting spin-singlet interlayer excitons which the type-i potential trap acts as a loss channel. I liked how thorough the paper was, there were lots of additional supplemental data plus relevant sources that collaborated stated assumptions and conclusions. I disliked how confusing some parts of the paper were which could be improved upon. <

We thank the referee for a careful reading of our manuscript and for acknowledging the systematic measurements we performed. We have improved upon the presentation in parts of the manuscript. We hope that these changes have improved our manuscript.

> Major Changes: 1.The outline/flow of the section “Trapped interlayer excitons” needs some work. The discussion for the type-i IX is not discussed until the very end of the section which is very odd and does not flow well in the paper. The discussion for these IX are not dependent on the data shown in figure 3 but rely on some of the supplementary data and figure 2 so it would be more natural to talk about where you think these potential traps are coming from for type-i before talking about type-ii. If you want to talk about the assignment of where the potential traps are coming from for both types at the same time, i.e. the end of this section, then I would change the way that this is done. Talk about all the relevant data including the lifetime data and then paint a clear picture of how these traps are different and where you think they are arising from. <

We thank the referee for the suggestions that we have included in the revised manuscript.

» We have now rearranged the argumentation in the part on “Trapped interlayer excitons”, where we give an explanation of the origin of the type-i emission feature at the end of the section.

> 2.You should add some word labels to figure 4d. It is clear what this is trying to illustrate to someone who reads the entire paper, but some people only look at the figures. Also figures 2e, 3c, and 4d are very similar and 2e is not even talked about in the paper. So, I am wondering if all three of these need to be included, or if you can combine 2e and 4d for when you talk about the potential trap origins are the end of the section. <

We thank the referee for the suggestions and have changed the manuscript accordingly.

» To make Figure 4 more clear, we added labels to panels 4c and 4d, see below. We have revised our manuscript as mentioned in the previous point. We combined Fig. 2e with Fig. 4d.

> **3. Figure 3b, the power log plot.** The blue data seems like two different power regimes instead of only one and the linear line seems to be only following the high-power points and not the low-power points, was this on purpose? For the 0.066uW power, some of the peaks are completely gone so how do you have a value for this power for that peak? If you did not include the 0.066uW data, what power range did you include? I count 8 points on the graph, but only six powers for figure 3a. <

We note that the 'two power regimes' of blue data are due to the weak PL signal of T-IX at powers below 1.32 uW. We integrate the PL intensity for the marked peaks with an energy range of ~ 3 meV (mentioned in the caption of Fig. 3). Therefore, we still obtain an integrated PL intensity even though the peaks are not resolvable, which results in the deviation of the blue data. Regarding the type-ii emission, the larger slope at low pump powers is due to the saturation behavior, which is one of the typical features of localized emitters.

» We now make the data in Fig. 3a and 3b consistent, and the revised version is shown below and updated in the revised manuscript. The power range for both figures is 0.066 μ W to 52.8 μ W. Additionally, we added to the caption of Fig. 3: "We note that the deviation of the integrated PL intensities of T-IX from linear dependence at pump power below 1.32 μ W is due to the poor signal-to-noise ratio".

> Minor Changes: 1. Figure 1a, some of the text is hard to read, some maybe try some other colors. The red-dash line is also not connected, fully outlining the HS section. <

We thank the referee for these suggestions.

>> Now we make the red-dash line more visible by slightly shifting it. Additionally, we change the position of 'MoSe₂' to get a better contrast and added a scale bar to the figure.

> 2. Figure 4c, maybe try to make it more obvious that this graph's y-axis is alpha like by adding a line where $\alpha = 1$ and $\alpha = 1.8$ is. <

>> We added the two lines into Fig. 4c and also marked the temperature where alpha

reaches a maximum value.

> 3.Supplemental 1 says “twist angle is measured 10 times”. What does this mean? Is this done by the same person? If the angle is estimated by examining the optical pictures, then just say so. The edges of each TMD are fairly obvious, so I am not sure what you mean by it was measured 10 times. That sounds like you used a program or instrument to measure the twist angle, but if it was done by optical inspection then that is an estimation even if it is done 10 times. Especially if it is done by the same person because they can just draw the same lines each time. Or did you use ten different pictures that the twist angle was estimated from? The SHG data makes it clear that the device is H-stacked instead of R-stacked, so the precise angle does not seem relevant here and getting a precise angle by human eyes from a single picture is hard. <

We measure the twist angle according to the optical contrast of the straight edges and the measurement is done by the same person. We zoom in on the optical image to draw the straight lines and in this case, the edges become blurred, which yields a certain range of measured twist angles. Hence, we measure it several times to get a statistical error. We agree with the referee that it is hard to state an uncertainty from examining the image multiple times. Rather, we now estimate the uncertainty to be $\pm 0.6^\circ$ by the largest (58.6°) and smallest (57.4°) twist angle determined by the pixelated edges, which give a resolvable contrast.

>> We now revised the caption of Fig. S1a to ”Optical image of the stacked MoSe₂/WSe₂

heterobilayer on PDMS. The twist angle is measured according to the straight edges and determined to be $58^\circ \pm 0.6^\circ$, where the uncertainty is given by the range of twist angles due to pixelated edges with a resolvable contrast.”

> **General Questions: 1.How many devices of this structure were measured with collaborating data? The supplemental data shows one other device, but that device does not have the type-ii PL emission.** <

We include data from two devices in our work. We show the data for the device with a smaller twist angle in the SI. For that device, we do not observe type-ii PL emission, which is possibly due to the atomic reconstruction occurring in the small twisted heterobilayer. The reconstruction could remarkably enlarge the cell size (A. Weston, et al., *Nat. Nanotechnol.* 15, 592–597 (2020)), resulting in smaller energy spacing between quantized energy levels and the difficulty for exciton localization at elevated temperatures.

Type-i emission that is related to defects is easier to observe owing to their insensitivity to the twist angle. This was also observed in previous work (K. L. Seyler, et al., *Nature* 567, 66–70 (2019)).

> **2.How did you deal with the PDMS residue for fabrication?** <

We did not use special treatment for PDMS residue. Regarding our fabrication method, we prepared MoSe₂ and WSe₂ monolayers on a PDMS stamp and placed one of them onto the other one. In this case, the interface between the monolayers will be residue-free. Additionally, the PDMS residue was reported to be an additional thin layer of material (A. Jain, *Nanotechnology* 29, 265203 (2018)), which is not expected to contribute to the type-i and type-ii features. On the other hand, the localized emitters without hBN encapsulation typically show clear spectral wandering (hundreds of μeV , A. Srivastava, et al., *Nat. Nanotechnol.* 10, 491–496 (2015), R. S. Daveau, et al., *APL Photonics* 5, 096105 (2020)). In our case, we measured the type-i IX with exposure time up to 20 seconds and still got a linewidth of $\sim 100\mu\text{eV}$ (see the right panel of Fig. 2a). Those narrow emission lines reflect that spectral wandering of the emitters is tiny, which is consistent with previous work (R. S. Daveau, et al., *APL Photonics* 5, 096105 (2020)) about single photon emitters in hBN encapsulated WSe₂ monolayers. Therefore, we expect that the heterobilayer is well encapsulated by the hBN.

» We added to the Methods section of our manuscript the following: ”We prepared MoSe₂ and WSe₂ monolayers on a PDMS stamp and placed them directly on top of each other, which guarantees that the interface between the monolayers will be clean.”

> **3.Do you have polarization data for the type-i localized IXs similar to figure S8? I know that the polarization is shown for the T and S peaks, but maybe also add the polarization for the low power type-i peaks to show that they are indeed also co-polarized.** <

Yes, the polarization data for the type-i localized IXs is shown in Fig. 2b.

***** Addressing suggestions in the attached scanned pdf file of one of the Reviewers *****

We thank the Reviewer for giving suggestions in the commented pdf file on how to improve the figures in our manuscript. We have incorporated these suggestions in the revised version of our manuscript. Here, we also answer the questions shown in the commented pdf file, but not given before.

> **1. How do you exclude type-ii emissions from defects?** <

The derived potential depth of the type-ii potential (~ 27 meV) is significantly larger than the reported defect potential (< 10 meV) in monolayer WSe₂ (Y. He, et al., *Optics Express* **24**, 8066-8073 (2016)) and MoSe₂ (C. Chakraborty, et al., *Optical Materials Express* **6**, 2081-2087 (2016)). On the other hand, the exciton-filling behavior with increasing pump power can be well-explained by the physical model based on quantized energy levels within moiré potential.

≫ We added an additional statement to the main manuscript.

> **2. What is the chemical treatment after the 1165 remover?** <

After using the 1165 remover, we used acetone and isopropyl alcohol.

≫ We added an additional statement to the Methods section of our manuscript.

> **3. Regarding Fig. S3, conclude that the emission lines come from defect-bound excitons.** <

≫ We added an additional statement to the caption of Fig. S3.

Reviewers' Comments:

Reviewer #1:

Remarks to the Author:

The authors basically addressed my questions, and if the other reviewers do not have more questions, I agree with its publication in Nature Communications.

Reviewer #2:

Remarks to the Author:

The authors have satisfactorily replied to the questions I raised in my first report, and I do think this manuscript is a good fit for publication in Nature Communications. I just have two further comments:

1) Regarding the first point from my previous report, I think it would be helpful to specify that the polarization behavior the authors observe is expected specifically for H-stacked bilayers. So that the trapped type-i interlayer excitons are of T-type specifically in the H-stacked system.

2) Regarding the addition of Supplementary Note 2, I think the expression for the moire length given is valid for (i) values of theta corresponding to the difference between the system angle and either 0 or 60 degrees and (ii) for theta expressed in radians. That is, by my calculations the moire length value of 9.4 nm that the authors report is obtained for theta = 2 degrees (expressed in radians) but not for 58 degrees. If the authors agree, it would be helpful to clarify these points in the Supplementary Note.

Reviewer #3:

Remarks to the Author:

Comments on Edits: I really appreciate that all the time and work that I put into annotating the paper was used well by the authors to improve the readability and quality of their paper. I think that the authors did a very good job with taking the questions and concerns from the reviewers and made the necessary changes. I think the readability of the trapped interlayer excitons section is much better and will make it easier for readers to understand.

I have a quick comment on my first question from the first review; I was curious if there were any other devices measured besides the two mentioned in the paper. It would make the paper stronger to mention that these results have been shown to be repeatable across # of devices showing the type-i, type-ii trapped interlayer excitons and the formation of the biexciton. If there is no other devices as of yet, that is fine, it was just a thought of whether this data has been reproduced by your group with another unreconstructed H-stacked device.

The final results of this paper are noteworthy and further explore the behavior of interlayer excitons in H-stacked moiré systems in various power and temperature regimes. By understanding interlayer excitons better within this system, future work can be done to exploit their behavior and properties in things like quantum emitter arrays.

***** Conventions *****

We answer the referees' criticism point-by-point. To this end, we keep the order of the referees' comments and mark it > **in blue and bold font** <. Our answers are in black and normal text. For each point, we indicate which parts of the manuscript have been changed >> in black in response to the referee's comments.

When we refer to Figures, Tables, Citations etc., we use the numbering of the revised version of the manuscript and supplemental material.

We add a marked-up version of our revised manuscript, where **added text is in blue, normal text, and removed text is in red, strikethrough text.**

***** Addressing Report of Reviewer 1 *****

> **The authors basically addressed my questions, and if the other reviewers do not have more questions, I agree with its publication in Nature Communications.** <

We thank the Reviewer for the support to publish our revised work in Nature Communications.

***** Addressing Report of Reviewer 2 *****

> **Regarding the first point from my previous report, I think it would be helpful to specify that the polarization behavior the authors observe is expected specifically for H-stacked bilayers. So that the trapped type-i interlayer excitons are of T-type specifically in the H-stacked system.** <

We stated that our studied heterobilayers are H-stacked (lines 87-88).

>> Additionally, we amended our manuscript and now explicitly write that the valley polarization behavior that we observe is expected from H-type heterobilayers (lines 161-164 and 178-179).

> **Regarding the addition of Supplementary Note 2, I think the expression for the moire length given is valid for (i) values of theta corresponding to the difference between the system angle and either 0 or 60 degrees and (ii) for theta expressed in radians. That is, by my calculations the moire length value of 9.4 nm that the authors report is obtained for theta = 2 degrees (expressed**

in radians) but not for 58 degrees. If the authors agree, it would be helpful to clarify these points in the Supplementary Note. <

We thank the referee for this clarification.

» We have now added the following paragraph to Supplementary Note 2: "We note that the relative angle between the two lattices is in the range of 0° to 30° due to the symmetry of the crystal (Pasqual Rivera, et al., Nat. Nanotechnol. 13, 1004–1015 (2018)). Therefore, a 58° twist angle corresponds to a relative angle of 2° between the layers, as is used in the calculation. A twist angle larger than 30° indicates the stacking type of the heterobilayer, which is H-type in our case."

***** Addressing Report of Reviewer 3 *****

> **Comments on Edits:** I really appreciate that all the time and work that I put into annotating the paper was used well by the authors to improve the readability and quality of their paper. I think that the authors did a very good job with taking the questions and concerns from the reviewers and made the necessary changes. I think the readability of the trapped interlayer excitons section is much better and will make it easier for readers to understand. <

We thank the Reviewer for the positive feedback on the changes we have made to the manuscript.

> **I have a quick comment on my first question from the first review; I was curious if there were any other devices measured besides the two mentioned in the paper. It would make the paper stronger to mention that these results have been shown to be repeatable across # of devices showing the type-i, type-ii trapped interlayer excitons and the formation of the biexciton. If there is no other devices as of yet, that is fine, it was just a thought of whether this data has been reproduced by your group with another unreconstructed H-stacked device.** <

We actually fabricated plenty of heterobilayers. However, we only choose the high-quality ones (that are the two samples presented in our work) for hBN encapsulation. We consider a sample of being of high quality when the heterobilayer is almost bubble-free under an optical microscope and also shows a good interlayer coupling evidenced by PL measurements at room temperature.

In the future, it is critical to improve the fabrication yield of high-quality heterobilayers to conduct systematic studies of the various excitonic states in heterobilayer systems.

» We added a small note on the fabrication process to the Methods section of our manuscript.

> **The final results of this paper are noteworthy and further explore the behavior of interlayer excitons in H-stacked moiré systems in various power and temperature regimes. By understanding interlayer excitons better within this**

system, future work can be done to exploit their behavior and properties in things like quantum emitter arrays. <

We thank the referee for expressing their appreciation of our work.

***** **Additional changes** *****

We uploaded the data that support our findings to the open repository Zenodo. Our data is now available at [10.5281/zenodo.10018495](https://doi.org/10.5281/zenodo.10018495). We have amended our data statement accordingly.